# Older Age Does Not Predict Inadequate Pain Management in Cancer Patients: A Multicenter Prospective Analysis from Italian Radiotherapy Departments (ARISE-Study)

**DOI:** 10.3390/cancers17183073

**Published:** 2025-09-19

**Authors:** Costanza M. Donati, Erika Galietta, Francesco Cellini, Arina A. Zamfir, Alessia Di Rito, Maurizio Portaluri, Anna Santacaterina, Filippo Mammini, Rossella Di Franco, Salvatore Parisi, Antonella Bianculli, Pierpaolo Ziccarelli, Luigi Ziccarelli, Domenico Genovesi, Luciana Caravatta, Francesco Deodato, Gabriella Macchia, Francesco Fiorica, Silvia Cammelli, Milly Buwenge, Lucia Angelini, Romina Rossi, Marco C. Maltoni, Nam P. Nguyen, Alessio G. Morganti, Savino Cilla

**Affiliations:** 1Radiation Oncology, IRCCS Azienda Ospedaliero-Universitaria di Bologna, 40138 Bologna, Italy; costanzamaria.donati@unibo.it (C.M.D.); erika.galietta@unibo.it (E.G.); filippo.mammini@studio.unibo.it (F.M.); silvia.cammelli2@unibo.it (S.C.); alessio.morganti2@unibo.it (A.G.M.); 2Department of Medical and Surgical Sciences (DIMEC), Alma Mater Studiorum University of Bologna, 40138 Bologna, Italy; milly.buwenge2@unibo.it (M.B.); romina.rossi10@unibo.it (R.R.); 3Dipartimento di Diagnostica per Immagini, Radioterapia Oncologica ed Ematologia, Fondazione Policlinico Universitario “A. Gemelli” IRCCS, 00168 Rome, Italy; francesco.cellini@policlinicogemelli.it; 4Dipartimento Universitario Diagnostica per Immagini, Radioterapia Oncologica ed Ematologia, Università Cattolica del Sacro Cuore, 00168 Rome, Italy; 5Radiotherapy Unit, IRCCS Istituto Tumori ‘Giovanni Paolo II’ Bari, 70124 Bari, Italy; alessia.dirito@aslbat.it; 6ASST Papa Giovanni XXIII, Piazza OMS 1, 24127 Bergamo, Italy; mportaluri@asst-pg23.it; 7U.O. di Radioterapia AOOR Papardo Piemonte, 98121 Messina, Italy; anna.santacaterina@virgilio.it; 8Department of Radiation Oncology, Istituto Nazionale Tumori-IRCCS-Fondazione G. Pascale, 80131 Napoli, Italy; r.difranco@istitutotumori.na.it; 9Radioterapia Opera di S. Pio da Pietralcina, Casa Sollievo della Sofferenza, 71013 Foggia, Italy; s.parisi@operapadrepio.it; 10Medical Physics Department, IRCCS-CROB—Centro di Riferimento Oncologico della Basilicata, 85028 Potenza, Italy; antonella.bianculli@crob.it; 11U.O. Radioterapia Oncologica, S.O. Mariano Santo, 87100 Cosenza, Italy; pziccarelli@virgilio.it (P.Z.); lziccarelli@virgilio.it (L.Z.); 12Radiation Oncology Unit, SS Annunziata Hospital, G. D’Annunzio University of Chieti-Pescara, 66100 Chieti, Italy; d.genovesi@unich.it; 13Department of Neuroscience, Imaging and Clinical Sciences, “G. D’Annunzio” University of Chieti-Pescara, 66013 Chieti, Italy; 14Radiation Oncology Unit, “C. e G. Mazzoni” Hospital, 63100 Ascoli Piceno, Italy; luciana.caravatta@sanita.marche.it; 15Radiotherapy Unit, Responsible Research Hospital, 86100 Campobasso, Italy; francesco.deodato@unicatt.it (F.D.); gabriella.macchia@responsible.hospital (G.M.); 16Istituto di Radiologia, Università Cattolica S. Cuore, 00168 Rome, Italy; 17U.O.C.di Radioterapia e Medicina Nucleare, Ospedale Mater Salutis di Legnago, 37045 Verona, Italy; francesco.fiorica@aulss9.veneto.it; 18Palliative Care Unit, IRCCS Istituto Romagnolo per lo Studio dei Tumori (IRST) “Dino Amadori”, 47014 orlì-Cesena, Italy; lucia.angelini@irst.emr.it; 19Medical Oncology Unit, Department of Medical and Surgical Sciences (DIMEC), University of Bologna, 40138 Bologna, Italy; marcocesare.maltoni@unibo.it; 20Department of Radiation Oncology, Howard University, Washington, DC 20060, USA; nam.nguyen@howard.edu; 21Medical Physics Unit, Responsible Research Hospital, 86100 Campobasso, Italy; savino.cilla@responsible.hospital

**Keywords:** pain management, radiotherapy, cancer, pain management index, ageing, LASSO, CART, geographic disparities, undertreatment

## Abstract

Many people receiving radiotherapy for cancer still live with poorly controlled pain. Older adults are often thought to be at greatest risk, yet this belief has rarely been tested in large, real-world groups. We followed more than two thousand patients attending thirteen radiotherapy centers across Italy and asked every person to rate their pain and list the medicines they were actually taking. We then compared the strength of each medicine with the reported pain level. Overall, four out of ten patients were undertreated, but being over 65 years old did not make this more likely. Instead, three other factors mattered most: pain that doctors classified as “not caused by the cancer”, treatment given with curative rather than palliative intent, and residence in central or southern Italy. These results suggest that improving pain care should focus on how pain is labeled, when treatment goals are curative, and where regional resources are limited.

## 1. Introduction

Pain is a common and distressing symptom in cancer patients, significantly impacting their quality of life. Recognized as a priority by the National Cancer Institute, its proper assessment and management are essential [1]. A substantial proportion of patients experience nociceptive or neuropathic pain throughout their disease course [2,3], contributing not only to physical suffering but also to emotional distress, which further reduces quality of life (QoL) [4,5,6,7]. Effective pain control is crucial, as it enhances functional status and autonomy [8,9].

Despite established guidelines and available analgesic options, pain remains undertreated [10,11,12,13,14]. Its psychosocial consequences, including increased psychological distress, depression, anxiety, and social isolation, highlight the need for a comprehensive, multidimensional management approach [15,16]. Additionally, disparities in pain treatment persist, influenced by factors such as age, gender, socioeconomic status, and cultural differences, underscoring the need for targeted interventions to ensure equitable care [17].

Particularly, managing cancer-related pain in older adults presents unique challenges compared to younger patients. These difficulties stem from age-related physiological changes, multiple coexisting health conditions, and concerns about medication side effects. Pain is often undertreated in this population due to fears of opioid use, heightened sensitivity to medications, and atypical pain presentations [18,19]. Aging affects drug metabolism and distribution, necessitating cautious opioid dosing to minimize adverse effects. Additionally, comorbidities complicate treatment by increasing the risk of drug interactions and limiting available options [20]. Cognitive impairment or the belief that pain is an inevitable part of aging can further contribute to underreporting and inadequate treatment. Moreover, in older adults, cancer pain management is further complicated by geriatric syndromes and polypharmacy, which call for systematic screening and individualized, multimodal care. Recent international guidance emphasizes offering opioids when indicated with risk-mitigation strategies and careful dose titration, alongside non-pharmacologic and interventional options, particularly pertinent to older patients. Contemporary reviews and practice resources also highlight that geriatric patients are at heightened risk of undertreatment and benefit from consistent use of patient-reported pain measures [21,22,23,24].

However, our previous analyses did not identify poorer pain management in elderly patients [25,26]. In contrast, when applying advanced statistical methods, we observed better pain management in a specific subgroup of elderly patients with breast cancer and pain of non-neoplastic origin [27]. To further explore this issue, we conducted an additional analysis stratifying patients into three age groups: young adults, middle-aged individuals, and elderly patients. This report aims to present the findings of that analysis.

## 2. Materials and Methods

### 2.1. Study Aims

The primary objective of this study was to assess the adequacy of pain management in older patients receiving treatment in radiotherapy (RT) departments. A secondary objective was to explore whether the impact of various predictive factors (such as gender, performance status, timing of evaluation, radiotherapy intent, primary tumor type, disease stage, pain characteristics, and the geographic location of the RT center) on pain management adequacy differed between older and younger patients.

### 2.2. Study Design

This was a pragmatic, multicenter, prospective cross-sectional study with consecutive enrollment over a predefined window. The sample size reflected feasibility across 13 centers; a formal a priori power calculation for between-age differences in the PMI was not performed. Patients were enrolled after providing informed consent, and the study received approval from the Ethics Committees of the participating centers (ARISE 327/2017/O/Oss). Eligible patients included those undergoing medical evaluation at the participating centers during the enrollment period. Data collection occurred in October–November 2019, i.e., prior to the COVID-19 pandemic. Thirteen Radiotherapy Departments participated (by macro-area: north = 3 departments (including 1 cancer center), 258 patients; center = 3 departments (including 1 cancer center), 157 patients; south = 7 departments (including 2 cancer centers), 938 patients). Participants were assessed regardless of whether their visit was part of an ongoing RT session or a post-treatment evaluation, with each patient being included in the study only once. During the clinical visit, data were collected using a structured case report form. The variables recorded included demographic and clinical information such as gender, age, Eastern Cooperative Oncology Group Performance Status (ECOG-PS), RT intent, primary cancer site, tumor stage, pain intensity (measured using the numeric rating scale, NRS), analgesic use, and pain classification (cancer-related pain, non-cancer pain, or mixed pain). However, we did not collect detailed information on the irradiated target structure/organ, treatment volumes, or fractionation beyond these descriptors. Moreover, the CRF did not include radiotherapy adherence variables (e.g., completion of the planned schedule, unplanned interruptions/dose reductions, or treatment drop-out); these were therefore not available for analysis. More generally, to ensure feasibility and consistent data quality across 13 centers with consecutive enrollment, we adopted a minimal, standardized case-report form focused on core variables aligned with the study aim (pain classification and intensity, analgesic regimen to derive the Pain Management Index (PMI), and key RT/tumor descriptors). Age was categorized a priori as 18–35 (young adults), 36–64 (middle-aged adults), and ≥65 years (older adults). We refer to 18–35 as ‘younger adults’ for clarity. While Adolescents and Young Adults (AYA) definitions vary internationally (e.g., 15–39 or 18–39 years), an 18–35 framework is also used in national AYA programs and peer-reviewed studies (e.g., the Dutch AYA ‘Young & Cancer’ Care Network), supporting our pragmatic choice. Moreover, only patients with pain scored as NRS ≥ 1 were included.

### 2.3. Inclusion Criteria

Patients were eligible for inclusion if they had a cancer diagnosis, regardless of primary tumor type, disease stage, or RT intent, and were receiving treatment in RT departments. Participants had to be at least 18 years old. Individuals with psychiatric conditions or neurosensory impairments that could interfere with data collection or informed consent were excluded.

### 2.4. End Points

Pain severity was categorized using the numeric rating scale (NRS): 0 for no pain (NRS: 0), 1 for mild pain (NRS: 1–4), 2 for moderate pain (NRS: 5–6), and 3 for severe pain (NRS: 7–10). The NRS is a widely used one-item measure with strong evidence of validity, reliability, and responsiveness in adults (including oncology) [28,29,30,31]. Analgesic treatment was also classified based on the type of medication prescribed: no analgesics (score 0), non-opioid analgesics (score 1), “weak” opioids (score 2), and “strong” opioids (score 3) Analgesic exposure was coded as analgesic class (0 = none; 1 = non-opioid; 2 = weak opioid; 3 = strong opioid), following the WHO analgesic ladder framework [32]; this is an objective classification based on recorded medications and is expected to have high inter-rater reliability under standardized data capture. The adequacy of pain management was assessed using the PMI, calculated by subtracting the pain score from the analgesic score. Negative PMI values indicated inadequate pain management [33]. Because of the cross-sectional design, each patient contributed a single PMI measurement, which offers a snapshot of analgesic adequacy and cannot capture within-patient fluctuations in pain and prescribing during or after RT. Moreover, we lacked granular data on the irradiated target structure/organ and fractionation parameters. Although RT intent and visit timing were recorded and considered in the analyses, residual confounding by unmeasured treatment factors cannot be excluded; therefore, results should be interpreted with appropriate caution. The PMI was used as a pragmatic indicator of analgesic adequacy. We recognize that the PMI does not account for dose, co-analgesics, or clinical response; it is intended for population-level quality assessment. Pain classification was recorded by the treating radiation oncologist as: cancer pain (attributable to the tumor) versus non-cancer pain (pre-existing/comorbid pain deemed not attributable to the cancer, including radiotherapy-related toxicities).

### 2.5. Statistical Methods

Categorical data are reported with frequencies and percentages. Fisher’s exact test for categorical variables was used to determine the *p*-values for statistical significance among different groups of patients.

In addition to conventional statistical analysis, because new and complex statistical methods based on machine learning have the potential to unravel hidden structures and uncover complex patterns in large databases, we performed the following supervised machine learning approach.

To identify predictive variables and develop predictive models, we applied the Least Absolute Shrinkage and Selection Operator (LASSO) algorithm alongside Classification and Regression Tree (CART) analysis. LASSO was utilized to select the most relevant variables for model inclusion, while CART facilitated the construction of decision tree-based models. To ensure model robustness, we performed 5-fold cross-validation, repeated 100 times. The predictive performance of the models was assessed using receiver operating characteristic (ROC) curves and quantified through the area under the curve (AUC) statistics. Additional details regarding the statistical analysis, model development, and software used are available in our previously published work [27]. All statistical analysis, including Lasso and machine learning training and internal validation, was performed using the XLSTAT 2022.1 and glmnet statistical packages v.4.1.3 (Addinsoft, New York, NY, USA. Analyses were exploratory and estimation-focused, emphasizing variable selection and pattern discovery (LASSO, CART) rather than hypothesis testing for a prespecified age-group contrast.

## 3. Results

A total of 2104 patients were enrolled in the study, among whom 1353 reported experiencing pain (NRS ≥ 1). Enrollment was uneven across macro-areas, with 7 Departments in the South contributing 938/1353 (69.3%) patients, versus 3 Departments each in the North (*n* = 272) and Center (*n* = 168).

Table 1 reports the comparison between the characteristics of all patients’ metrics, with PMI < 0 and PMI ≥ 0.

### 3.1. Selection of Variables

The LASSO regression analysis identified several variables as significant determinants of a PMI score less than 0, indicative of suboptimal pain management. In order of importance (based on the magnitude of LASSO coefficient), these variables are: the type of pain, the type of cancer (breast cancer vs. all other cancers), the aim of therapy (curative vs. palliative radiotherapy), the timing of visit (during or at the end of therapy), the location of the centers (north, central of south Italy), the tumor stage (metastatic vs. non-metastatic), the performance status and the age.

Using these selected variables, four CART classifiers for different patient subgroups were trained and developed as follows. Model A: predictive model for inadequate pain management in all patients, classified in three age groups: 18–35, 36–64, and ≥65 years; model B: predictive model for inadequate pain management in all patients, classified in two groups: 18–64 and ≥65 years; Model C: predictive model for inadequate pain management (only patients aged ≥ 65 years included); and Model D: predictive model for inadequate pain management (only patients aged 18–64 years included).

### 3.2. Model Performance

The performances of the four CART models were evaluated using ROC and AUC values, as reported in Figure 1. The mean AUCs were 0.744 ± 0.010, 0.740 ± 0.010, 0.791 ± 0.020, and 0.767 ± 0.015, for the Models A, B, C, and D, respectively.

### 3.3. Model A: Predictive Model for Inadequate Pain Management in All Patients (Three Groups)

In this model, patients were stratified by age into three groups: young adults (18–35 years), middle-aged adults (36–64 years), and older adults (≥65 years). The predictive model included all 1353 patients reporting NRS pain scores ≥ 1 (Figure 2). At the initial division, inadequate analgesia (PMI < 0) was substantially higher for patients classified as having non-cancer pain (74.3%) compared to cancer pain (34.2%). Within the cancer pain subgroup, analgesic inadequacy was significantly higher in patients receiving curative RT (49.4%) compared to palliative RT (28.8%). For non-cancer pain, undertreatment was highest in patients with breast cancer (83.8%) versus other cancers (65.5%). Age stratification emerged in a deeper branch for breast cancer pain not labeled as cancer-related and treated in RT centers in central and southern Italy: inadequacy was slightly higher in middle-aged adults (89.5%) compared to other ages (84.0%).

### 3.4. Model B: Predictive Model for Inadequate Pain Management in All Patients (Two Groups)

In this model, patients were stratified by age into two groups: young/middle-aged adults (18–64 years) and older adults (≥65 years). In patients aged 18–64 and ≥65 years (Figure 3), the tree first addresses dichotomy, separating pain that clinicians attributed to cancer from pain considered non-cancer-related. In the cancer pain branch, 34.2% of patients were inadequately treated, whereas in the non-cancer branch, the proportion surged to 74.3%. Within the cancer pain arm, the next split was governed by the intent of RT. Patients receiving curative RT had an undertreatment rate of 49.4%, contrasted with 28.8% among the 671 patients treated palliatively. Turning to the non-cancer branch, the second-level partition distinguishes breast cancer from all other tumors. Breast cancer pain proved the most vulnerable category: 83.8% of the 210 patients in this subgroup had a negative PMI, compared with 65.5% of the 229 patients grouped as “other cancers.” In the breast cancer subgroup with non-cancer pain and undergoing treatment in central and southern Italy, undertreatment was slightly higher (89.0%) in younger/middle-aged adults compared to older adults (84.0%).

### 3.5. Model C: Predictive Model for Inadequate Pain Management (Only Patients Aged ≥ 65 Years Included)

In patients aged ≥ 65 years (Figure 4), undertreatment rates remained significantly higher in non-cancer pain (71.3%) than in cancer pain (35.6%). Among older patients with cancer pain, inadequacy was notably more frequent in those receiving curative radiotherapy (50.0%) compared to palliative radiotherapy (31.1%). Geographical differences further amplified this disparity; curative RT patients in central and southern Italy showed higher undertreatment (53.8%) compared to northern Italy (41.7%). Similarly, within the palliative RT subgroup, patients treated in central Italy exhibited the highest inadequacy rates (64.0%), while the northern regions reported markedly lower undertreatment rates (15.4%). Geographical differences also affected patients with non-cancer pain. Among older breast cancer patients whose pain was classified as non-cancer-related, inadequacy was notably higher in southern Italy (82.9%) compared to those treated in northern or central Italy (70.0%).

### 3.6. Model D: Predictive Model for Inadequate Pain Management (Only Patients Aged 18–64 Years Included)

For patients aged 18–64 years (Figure 5), the highest inadequacy rates were also observed in non-cancer pain (78.7%) compared to cancer pain (32.7%), particularly for breast cancer cases (88.1%). In the cancer pain subgroup, analgesic inadequacy was higher under curative (48.8%) compared to palliative RT (25.9%). Furthermore, breast cancer pain under curative RT exhibited an especially high undertreatment rate (80.0%) compared to other cancers (39.2%). Evaluation timing influenced analgesic inadequacy in several subgroups. For non-cancer breast pain, inadequacy rates remained extremely high both at the start (90.9%) and end (86.2%) of RT, but showed minimal improvement. Conversely, for non-cancer pain associated with other tumors, analgesic inadequacy decreased from 72.7% at the start to 55.6% by the end of treatment, indicating some improvement over time.

Finally, for readability, we include a concise age-stratified summary table that highlights primary, secondary, and context-dependent predictors of undertreatment (Table 2).

## 4. Discussion

This prospective, multicenter study examined the adequacy of pain control among patients attending Italian RT departments. Of 2104 consecutive patients screened, 1353 reported pain of at least mild intensity (numeric rating scale ≥ 1) and were included in the analysis. Analgesic adequacy was quantified with the PMI. To identify the most informative predictors, we first applied the LASSO, which selected a parsimonious set of variables. These variables were then entered into a CART to generate an interpretable decision model. Overall, 42% of patients had a negative PMI, indicating undertreatment. Undertreatment was markedly more common when pain was recorded as non-cancer-related (74.3%) than when it was attributed to cancer (34.2%). Within the cancer pain group, patients receiving curative RT were undertreated more often than those receiving palliative RT (49.4% vs. 28.8%). In the non-cancer category, breast cancer patients showed the highest rate of undertreatment (83.8%). Geographical variation was pronounced, with RT centers in central and southern Italy consistently reporting poorer pain control than those in the north. After adjustment for these factors, chronological age was not correlated with a clinically relevant difference in terms of inadequate analgesia. Moreover, the observed geographic gradient should be interpreted with caution because macro-areas contributed different numbers of Departments and patients, and we did not model center-level clustering. Thus, part of the signal may reflect center heterogeneity rather than geography per se. Future studies should employ multilevel models (center random effects) and/or equal-weight per-center summaries to separate macro-area from center effects.

### 4.1. Age-Specific Determinants of Pain Management Adequacy

Consistent with our earlier ARISE reports [25,26,27], pain documented as non-cancer-related and pain experienced by breast cancer patients showed the greatest probability of undertreatment. The aggregate proportion of negative PMI values did not differ appreciably between older patients (≥65 years) and those aged 18–64 years; however, the determinants of undertreatment diverged between these two age cohorts.

Older adults (≥65 years)

The single most influential variable was the geographic location of the RT facility. Across several decision-tree branches, the likelihood of a negative PMI was notably higher in centers located in the center or south of Italy than in those in the north. These geographic disparities are unlikely to stem from heterogeneity in RT protocols, which are nationally standardized. Rather, they plausibly reflect organizational differences in pain-care resources (e.g., limited clinician availability, time constraints, variable adherence to guidelines) that have been associated with undertreatment in Italian and European settings [33,34,35]. In addition, a more fatalistic worldview, documented in Mediterranean populations [36] and plausibly accentuated among older adults, may discourage patients in the south from openly reporting pain, thereby hindering appropriate dose escalation. Similar regional gradients in analgesic quality have been reported both within other countries [37] and in our prior analyses that combined cancer- and non-cancer-related pain [25,26,27].

Adults aged 18–64 years

No north–south differential emerged in this group. Instead, several subanalyses demonstrated a consistent improvement in PMI between the start and completion of RT. One interpretation is that younger individuals become progressively more comfortable communicating symptoms as treatment progresses, whereas many older patients, already accustomed to chronic osteo-articular discomfort, may adjust their internal pain benchmarks and underreport milder sensations, a phenomenon akin to “response-shift bias” [38]. Mixed-methods research that couples quantitative metrics with qualitative interviewing could clarify how age, expectations, and symptom appraisal intersect to shape pain reporting during RT.

### 4.2. Comparison with Previous Literature

Our finding that age alone was not a determinant of undertreatment aligns with the recent international literature, which stresses that geriatric assessment, proactive monitoring, and tailored analgesic choices, not age per se, may guide management in older adults with cancer. Incorporating patient-reported pain assessment and geriatric pharmacology principles may help narrow residual gaps in adequacy [21,22,23].

The pattern we observed, namely the absence of a clinically relevant age penalty in analgesic adequacy, contrasts with the findings of previous studies [39,40,41,42]. Each of those investigations reported either a direct association between older age and undertreatment or a significant age-related gradient in the PMI. Several contextual factors might explain why our Italian cohort deviates from this international trend.

First, population structure differs markedly. Italy currently has the oldest demographic profile in Europe, with almost one-quarter of residents aged ≥ 65 years and a median age exceeding 48 years [43]. More than 60% of newly diagnosed cancer cases in the country occur in this age bracket [44]. In such a setting, clinicians may view older adults as the ‘typical’ rather than ‘exceptional’ oncology patient, reducing the age-related therapeutic inertia reported elsewhere and narrowing the PMI gap between age groups.

Second, national policies introduced over the past decade, such as mandatory pain-score documentation (Italian Law 38/2010) and widespread palliative-care training, may have improved prescribing behavior uniformly across ages, thereby diluting the age signal still evident in historical series. Importantly, the two US studies [39,41] pre-date these initiatives by decades, and the Norwegian [42] and pan-European [40] nursing-home studies focused on settings with very different staffing models and reimbursement structures.

Finally, the age threshold that triggers caution in opioid titration may be higher in everyday Italian practice, with prescribers perhaps reserving the label “elderly” for those well beyond 75–80 years. Future analyses stratified by finer age bands could test this hypothesis and determine whether a treatment gap re-emerges among the oldest-old.

### 4.3. Strengths and Limitations

Strengths

This investigation represents, to our knowledge, the largest prospective, multicenter assessment of analgesic appropriateness in an RT cohort. The analytical pipeline combined LASSO, which performs simultaneous shrinkage and variable selection to minimize over-fitting, with CART, a method that converts the selected predictors into an intuitive decision structure and reveals higher-order interactions and clinically actionable cut-points. Because artificial intelligence, and in particular machine learning techniques, are increasingly used in the realm of healthcare for the development of predictive models, a balance between performance (discriminative accuracy) and interpretability is warranted. Many models are considered as black box models, i.e., models that are highly complex and not straightforwardly interpretable to humans. Lack of interpretability in predictive models can undermine trust in those models, especially in health care, in which so many decisions are, literally, life and death issues. Interpretable machine learning methods have the potential to provide the means to overcome some of these issues, but are largely unexplored within the care domain of pain management. Decision trees represent intrinsic interpretable models, without the need for post hoc explanations: the resulting tree structure allows users to understand and interpret the decision-making process easily, providing a simple scheme to analyze complex data and identify patterns and relationships. This aspect represents a major strength of the present investigation.

Limitations

First, the study interrogated pharmacological adequacy alone; it did not examine downstream outcomes such as symptom interference or health-related QoL. Second, each patient contributed a single PMI measurement obtained either during or at the close of RT, precluding a formal time-series analysis and obscuring the relative contribution of referring physicians versus radiation oncologists. Nevertheless, the significantly lower prevalence of negative PMI at treatment completion (38% vs. 47%) suggests that on-treatment dose titration and/or the antalgic effect of RT contributed to improved control.

Because each patient contributed a single-visit assessment, our data provide a snapshot of analgesic adequacy that cannot capture within-patient fluctuations in pain and prescribing during or shortly after RT. The timing variable (‘on-treatment’ vs. ‘end-of-treatment’) is coarse and does not encode the number of fractions received, cumulative dose, days since last fraction, or whether the visit occurred near the peak of acute symptoms. Although RT intent and visit timing were recorded and considered, residual confounding by unmeasured treatment-course factors cannot be excluded; therefore, timing-related contrasts should be interpreted with appropriate caution.

Another limitation is the absence of radiotherapy adherence data. We did not collect information on completion of the planned RT, unplanned interruptions or dose reductions, nor on treatment drop-out. While our cross-sectional design reduces the likelihood that attrition biased the PMI assessed at the clinical visit, these unmeasured variables could still confound the association between patient-/treatment-level factors and analgesic adequacy. Future studies should link clinical assessments with RT treatment logs to evaluate these effects explicitly.

Moreover, data were collected pre-pandemic. Since 2020, oncology services have undergone substantial changes, including telemedicine adoption [45] and radiotherapy pathway reconfigurations (e.g., broader use of hypofractionation) [46], with documented effects on patient experience and care processes [47]. These shifts may influence pain assessment and analgesic prescribing; thus, our findings should be interpreted as a pre-COVID baseline, with cautious generalizability to current practice.

A further limitation is the restricted generalizability of our findings. Although the study was multicentric in design, it was conducted entirely within a single national healthcare system. Consequently, the regional disparities observed in PMI, particularly the differences between northern and southern Italian centers, may reflect country-specific organizational, cultural, or infrastructural factors and should not be assumed to apply universally across other settings.

Another key limitation is the pragmatic breadth of our dataset. The CRF did not capture several clinically relevant domains (e.g., detailed analgesic dosing and titration, breakthrough pain, comorbidities/polypharmacy, and formal geriatric assessment, or radiotherapy adherence metrics such as interruptions/completion). Together with the cross-sectional design, this limits causal inference and leaves room for residual confounding. While our prespecified modeling strategy (LASSO for variable selection and CART for pattern discovery) improves transparency and reduces overfitting, it cannot compensate for unmeasured variables. Future work should prospectively link clinical assessments with RT treatment logs and richer geriatric/pain measures, with longitudinal follow-up to examine trajectories in analgesic adequacy.

The PMI itself carries inherent constraints. It relies on the historical dichotomy between “weak” and “strong” opioids [42], and its association with QoL metrics is inconsistent: a negative PMI is not invariably accompanied by an expressed need for intensified analgesia [48], although it does correlate with greater pain-related functional limitation [49]. Moreover, classic series [39,44,49,50,51] calculate the index from prescribed rather than consumed medication. We mitigated this bias by interviewing patients regarding actual intake. Even so, the PMI deemphasizes titration adequacy by assigning every strong-opioid regimen a non-negative score, irrespective of dose intensity or resultant relief [52]. Despite these imperfections, the index remains the most extensively utilized surrogate for pharmacological appropriateness because of its ease of computation and its reproducible linkage with global indicators of pain care [53]. Therefore, PMI < 0 should be interpreted as potential undertreatment, not proof of ineffective analgesia, and results have to be interpreted with appropriate caution. However, prior work supports its utility for benchmarking while also highlighting its limitations [14,50,54,55].

Another limitation is the absence of a prospective power analysis for detecting age-group differences in analgesic adequacy. Accordingly, the null association for age per se should be interpreted with caution, as small-to-moderate effects cannot be excluded (i.e., potential type II error). These results should be considered hypothesis-generating. Moreover, in radiotherapy settings, pain can fluctuate with target organ, treatment phase, and toxicity, underscoring the need for systematic assessment [56]. Our cross-sectional snapshot captures status at a single visit and cannot represent symptom trajectories during/after RT.

Finally, because the design was cross-sectional, each patient contributed a single PMI measurement obtained at the clinical visit (during or at the end of RT). Consequently, RT attrition during the course would not affect the adequacy score recorded at that visit. However, we did not collect dropout rates by region; therefore, while selection effects cannot be entirely excluded, they are unlikely to explain the observed PMI gradient.

Lastly, Lasso regression also presents some limitations, in particular its difficulty with highly correlated features, where it arbitrarily selects one predictor and excludes others, potentially leading to unstable or biased results. It can also produce a biased estimator because the penalty can shrink coefficients towards zero, causing the model to underfit the data if the penalty is too high. Furthermore, Lasso can be problematic for non-sparse datasets with many relevant features, as it will select only a subset of predictors by forcing many coefficients to zero.

### 4.4. Future Perspectives

Nearly one patient in two treated in Italian RT departments continues to receive suboptimal analgesia, a proportion that calls for concerted action. Three operational priorities emerge. First, routine documentation of the PMI alongside pain intensity would enable real-time identification of undertreated individuals. In Italy, systematic pain recording is mandated by national policy (Italian law 38/2010), but documentation alone does not always trigger timely action; using PMI < 0 as a simple “adequacy check” functions as a screen for potential undertreatment and prompts same-day review/titration, complementing, not replacing, existing pain tools. Second, targeted education for physicians and nursing staff, covering opioid titration, adjuvant use, and communication skills, could enhance both detection and management of pain. While such training is expected in routine care, evidence shows persistent knowledge gaps and variable guideline adherence, particularly in radiotherapy departments; brief refreshers (e.g., titration checklists, short e-learning) can close these gaps. [57]. Third, symptom control should be embedded within multidisciplinary pathways that integrate oncologists, palliative care specialists, pharmacists, and psychologists. Although multidisciplinary team pathways often exist on paper, several studies report limited bidirectional referral between radiotherapy and palliative care; pragmatic steps include appointing a pain/palliative liaison, setting automatic referral triggers (e.g., persistent PMI < 0 at two visits), and auditing undertreatment rates by center to reduce unwarranted variation [57]. Given our predictors, these actions should be prioritized for patients with non-cancer pain, those on curative-intent pathways, and in central/south macro-areas. Further research is warranted to refine these interventions. Prospective studies should chart the trajectory of pain intensity and PMI values throughout the RT course to pinpoint critical time-points for optimization. Cluster-randomized trials could test whether focused educational packages improve prescribing behavior and patient-reported outcomes. Linkage with radiotherapy delivery records (fractionation, interruptions, completion) would clarify whether adherence metrics influence analgesic adequacy trajectories. Finally, a granular characterization of pain syndromes in the RT population may help tailor consultation timing and content, ensuring that radiation oncologists address pain proactively rather than reactively.

## 5. Conclusions

In contrast with earlier international series, our multicenter analysis did not detect poorer analgesic adequacy in cancer patients aged ≥ 65 years. Therefore, age per se does not predispose to suboptimal pain control; however, the factors that govern analgesic adequacy differ across age groups. In older adults, center geography was the principal driver of undertreatment, whereas in patients aged 18–64 years, the timing of clinical assessment exerted the strongest influence. However, due to the study limitations, our findings should be interpreted as exploratory and hypothesis-generating, identifying organizational targets for improving pain-care processes rather than implying causality. Future longitudinal and interventional studies are needed to validate these age-dependent patterns and to develop targeted strategies that eliminate the remaining treatment gap. These studies should be prospectively powered to detect clinically meaningful between-age differences in PMI, and they should replicate and update these analyses in post-pandemic cohorts to confirm whether the observed patterns persist under current care models.

## Figures and Tables

**Figure 1 cancers-17-03073-f001:**
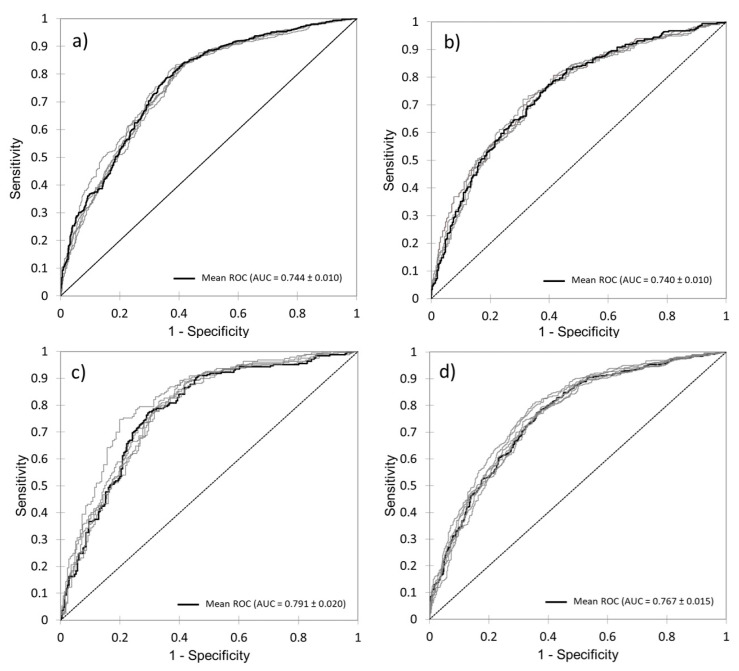
ROC curves and average AUC of the four CART models. K-fold cross-validation (k = 5) was used to estimate and compare the performance of different CART models. After five rounds of training/validation rotation, the average AUC was calculated: (**a**) Model A, (**b**) Model B, (**c**) Model C, and (**d**) Model D. CART: classification and regression tree; ROC curve: receiver operating characteristic curve; AUC: area under the curve.

**Figure 2 cancers-17-03073-f002:**
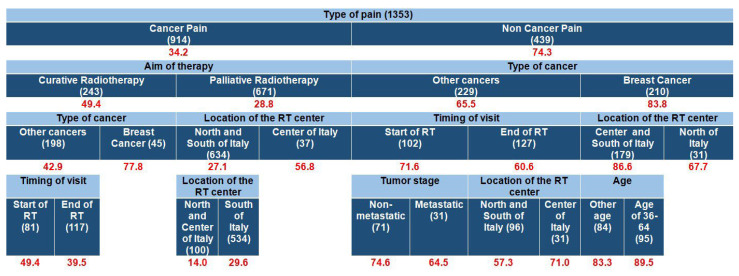
Predictive model for inadequate pain management: red numbers represent the proportion (%) of patients with inadequate pain management (PMI < 0); all (1353) patients with NRS ≥ 1 are included. The numbers in parentheses represent the patient count within each specific subgroup. In this model, patients were stratified by age into three groups: young adults (18–35 years), middle-aged adults (36–64 years), and older adults (≥65 years). Each row is a distinct subgroup defined by the labels shown in that row. When the table has multiple blocks/columns, read left to right: each block further splits the subgroup to its left, and the patient counts within a block sum to their parent. Rows are mutually exclusive and together cover all patients in the parent group. The model considered all candidate predictors listed in Methods; a variable appears only if selected by the algorithm.

**Figure 3 cancers-17-03073-f003:**
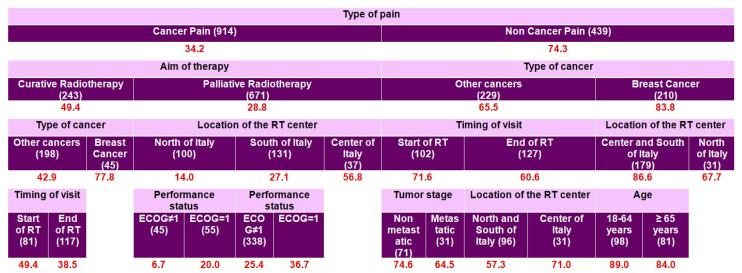
Predictive model for inadequate pain management: red numbers represent the proportion (%) of patients with inadequate pain management (PMI < 0); all (1353) patients with NRS ≥ 1 included. The numbers in parentheses represent the patient count within each specific subgroup. Patients were stratified by age into two groups: young/middle-aged adults (18–64 years) and older adults (≥65 years). Each row is a distinct subgroup defined by the labels shown in that row. When the table has multiple blocks/columns, read left to right: each block further splits the subgroup to its left, and the patient counts within a block sum to their parent. Rows are mutually exclusive and together cover all patients in the parent group. The model considered all candidate predictors listed in Methods; a variable appears only if selected by the algorithm.

**Figure 4 cancers-17-03073-f004:**
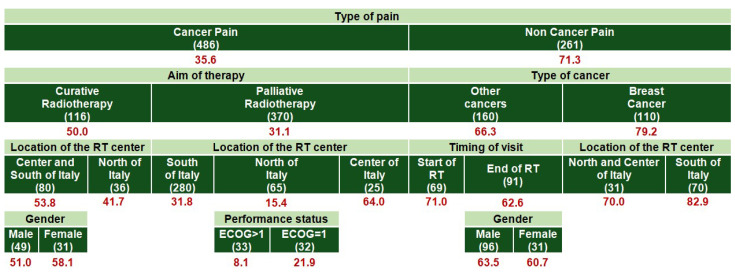
Predictive model for inadequate pain management: red numbers represent the proportion (%) of patients with inadequate pain management (PMI < 0); only 747 patients with NRS ≥ 1 and with age ≥ 65 years were included. The numbers in parentheses represent the patient count within each specific subgroup. Each row is a distinct subgroup defined by the labels shown in that row. When the table has multiple blocks/columns, read left to right: each block further splits the subgroup to its left, and the patient counts within a block sum to their parent. Rows are mutually exclusive and together cover all patients in the parent group. The model considered all candidate predictors listed in Methods; a variable appears only if selected by the algorithm.

**Figure 5 cancers-17-03073-f005:**
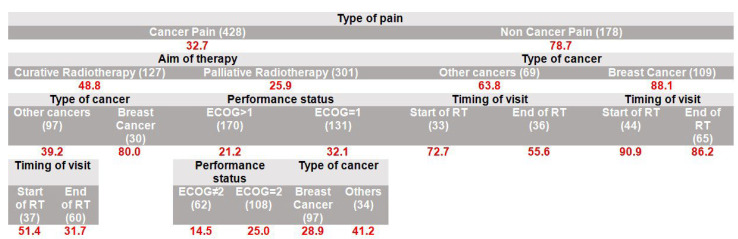
Predictive model for inadequate pain management: red numbers represent the proportion (%) of patients with inadequate pain management (PMI < 0); only 606 patients with NRS ≥ 1 and with age 18–64 years were included. The numbers in parentheses represent the patient count within each specific subgroup. Each row is a distinct subgroup defined by the labels shown in that row. When the table has multiple blocks/columns, read left to right: each block further splits the subgroup to its left, and the patient counts within a block sum to their parent. Rows are mutually exclusive and together cover all patients in the parent group. The model considered all candidate predictors listed in Methods; a variable appears only if selected by the algorithm.

**Table 1 cancers-17-03073-t001:** Patients characteristics (1353) and results of the univariate analysis.

	All (1353)		PMI < 0		PMI ≥ 0		
	Number (%)		Number (%)		Number (%)		*p*
Gender							<0.001
Male	639	47.2	257	40.2	382	53.5	
Female	714	52.8	382	59.8	332	46.5	
Age, years							0.780
18–35	27	2.0	12	1.9	15	2.1	
36–64	579	42.8	268	41.9	311	43.6	
≥65	747	55.2	359	56.2	388	54.3	
ECOG-PS							<0.001
1	831	61.4	468	73.2	363	50.8	
2	334	24.7	122	19.1	212	29.7	
3	160	11.8	45	7.0	115	16.1	
4	28	2.1	4	0.7	24	3.9	
Aim of treatment							<0.001
Curative	645	47.7	421	65.9	224	31.4	
Palliative	708	52.3	218	34.1	490	68.6	
Primary Tumor							<0.001
Breast	422	31.1	263	41.2	159	22.3	
Prostate	143	10.5	65	10.2	78	10.9	
Gastrointestinal	134	9.9	43	6.7	91	12.7	
Endometrial/Cervical	72	5.3	39	6.1	33	4.6	
Lung	191	14.1	58	9.1	133	18.6	
Head and Neck	117	8.6	62	9.7	55	7.7	
Others	274	20.2	109	17.1	165	23.1	
Tumor stage							<0.001
Metastatic	737	54.5	240	37.6	497	69.6	
Non-Metastatic	616	45.5	399	62.4	217	30.4	
Type of Pain							<0.001
Cancer pain or mixed pain	914	67.5	313	49.0	601	84.2	
Non-cancer Pain	439	32.5	326	51.0	113	15.8	
Pain score							<0.001
NRS: 1–4	591	43.7	231	36.2	360	50.4	
NRS: 5–6	509	37.6	274	42.8	235	32.9	
NRS: 7–10	253	18.7	134	21.0	119	16.7	
Analgesic score							<0.001
No therapy	327	24.2	327	51.1	0	0	
Analgesics	551	40.7	275	43.1	276	38.7	
Weak Opioids	194	14.3	37	5.8	157	22.0	
Strong Opioids	281	20.8	0	0	281	39.3	
Location of the radiotherapy center							<0.001
North of Italy	258	19.1	103	16.1	155	21.7	
Center of Italy	157	11.6	102	16.0	55	7.7	
South of Italy	938	69.3	434	67.9	504	70.6	
Timing of visit							0.040
During Therapy	728	53.8	325	50.9	403	56.4	
End of Therapy	625	46.2	314	49.1	311	43.6	

Legend: ECOG = Eastern Cooperative Oncology Group performance status, 0–4 scale: 0 fully active; 1 restricted in physically strenuous activity but ambulatory and able to carry out light work; 2 ambulatory and capable of all self-care but unable to carry out any work activities; up and about >50% of waking hours; 3 capable of only limited self-care; confined to bed or chair > 50% of waking hours; 4 completely disabled; NRS = numeric rating scale (0–10; 0 = no pain, 10 = worst imaginable pain); Percentages are rounded to one decimal; totals may differ from 100.0% by ±0.1 due to rounding.

**Table 2 cancers-17-03073-t002:** Age-stratified summary of predictors of undertreatment (PMI < 0). Ranking reflects consistency/strength across LASSO and CART: primary = most consistent signals; secondary = frequent signals; tertiary = context-dependent (appear in CART subsets). This table is descriptive and does not introduce new estimates.

Age Group	Primary	Secondary	Tertiary (Context-Dependent)
18–64 years	Type of pain	Aim of therapy	Type of cancer; Performance status
≥65 years	Type of pain	Aim of therapy	Type of cancer; Location of the RT center; Timing of the visit

Abbreviation: RT: radiotherapy.

## Data Availability

The data of this study are available from the corresponding author upon reasonable request to the corresponding author DOI: https://zenodo.org/records/16404475. Accessed on 24 July 2025.

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
