# Peer review of "Older Age Does Not Predict Inadequate Pain Management in Cancer Patients: A Multicenter Prospective Analysis from Italian Radiotherapy Departments (ARISE-Study)"

_cancers, 2025, doi:10.3390/cancers17183073_

Round 1
Reviewer 1 Report
Comments and Suggestions for Authors
From a biostats and clinical epidemiology point of view, here are some comments for the Authors:
- October and November 2019, enrollment is quite outdated, why?
- Centro di Riferimento Oncologico della Basilica, typo Basilicata
- central and southern Italy compared to the north, RT protocols have been properly standardized along the country, a so huge difference needs to be deeper elucidated (i.e. do older people in central and southern Italy show a RT protocol drop-out rate larger than in northern one? and so on); limited regional resources should not play any true role, at least in this topic
- when applying advanced statistical methods, I don't agree at all with you, stats methods must not change the trial results!
- an interesting covariate is lacking: has the pt completed his/her RT planned schedule? Any dose reduction? Any drop-out?
- the performance status, add ECOG, please
- four CART classifiers, what about to choose and report only one of them? My suggestion is to choose the most representative model and report/comment only that one (of course, including geographical locations and age as covariates/features)
- figure 2, mind that 34.2% + 74.3% does not add to 100%
- the same for figures 3,4,5
- table 1, cancer/non-cancer pain does not add to 100%
- no XAI explainable AI techniques have been used, to convert the black box into a glass one
Author Response
Comment 1: From a biostats and clinical epidemiology point of view, here are some comments for the Authors: October and November 2019, enrollment is quite outdated, why?
Response 1: We thank the Reviewer for this thoughtful observation. The present manuscript reports an additional sub-analysis of the multicenter ARISE study. Over time, our group has sequentially reported the main trial results (Donati CM et al., Cancers 2022) and three further sub-analyses (Donati CM et al., Cancers 2023; Donati CM et al., Frontiers in Oncology 2025; Donati CM et al., 2025 submitted). Given the multicenter nature of ARISE and the large number of collaborators involved, these analyses were completed and disseminated in stages. The current work addresses a distinct research question derived from the original dataset; therefore, the enrollment window (October–November 2019) reflects the parent trial period and is appropriate for this focused analysis.
Comment 2: Centro di Riferimento Oncologico della Basilica, typo Basilicata central and southern Italy compared to the north, RT protocols have been properly standardized along the country, a so huge difference needs to be deeper elucidated (i.e. do older people in central and southern Italy show a RT protocol drop-out rate larger than in northern one? and so on); limited regional resources should not play any true role, at least in this topic
Response 2: We thank the Reviewer for these important clarifications. We corrected the affiliation typo (“IRCCS-CROB—Centro di Riferimento Oncologico della Basilicata”). As for the North vs. Center/South difference, we agree that RT protocols are standardized nationally; however, our outcome (PMI) reflects pain assessment and analgesic prescribing, which are sensitive to organizational factors and resource constraints. Italian surveys and expert reports highlight persistent gaps in guideline adherence, pain assessment, and service organization, despite national regulations, which plausibly contribute to undertreatment variability (e.g., Cascella et al. 2022; Marinangeli et al. 2021). Broader evidence also shows high, and heterogeneous, rates of undertreatment by PMI in Europe/Italy (Deandrea et al. 2008; Greco et al. 2014). We have added citations to reflect this context. Regarding dropout: our cross-sectional design uses a single per-patient PMI at the clinical visit; therefore, RT attrition would not bias the adequacy score recorded at that visit, and we did not collect regional dropout. These points are now clarified in the manuscript.
Manuscript changes:
- Affiliation: “Basilica” → “Basilicata”.
- Discussion, added (section 4.1.): “These geographic disparities are unlikely to stem from heterogeneity in RT protocols, which are nationally standardized. Rather, they plausibly reflect organizational differences in pain-care resources (e.g., limited clinician availability, time constraints, variable adherence to guidelines) that have been associated with undertreatment in Italian and European settings [new references added].”
- Limitations design/dropout: added (section 4.3.): “Because the design was cross-sectional, each patient contributed a single PMI measurement obtained at the clinical visit (during or at the end of RT). Consequently, RT attrition during the course would not affect the adequacy score recorded at that visit. We did not collect dropout rates by region; therefore, while selection effects cannot be entirely excluded, they are unlikely to explain the observed PMI gradient.”
- Reference list, added:
Cascella M, Vittori A, Petrucci E, et al. Strengths and Weaknesses of Cancer Pain Management in Italy: Findings from a Nationwide SIAARTI Survey. Healthcare (Basel). 2022;10(3):441. doi:10.3390/healthcare10030441.
Marinangeli F, Saetta A, Lugini A, et al. Current management of cancer pain in Italy: Expert opinion paper. Open Med (Wars). 2021;17(1):34–45. doi:10.1515/med-2021-0393.
Othman WM, Al-Atiyyat NM. Knowledge, perceived barriers, and practices of oncology nurses regarding cancer pain management. Electron J Gen Med. 2022;19(6):em406.
Comment 3: when applying advanced statistical methods, I don't agree at all with you, stats methods must not change the trial results!
Response 3: We understand the rewievers’ concerns. However, real-world data is often complex and high-dimensional. Traditional approaches of data analysis are in most cases ineffective and can only model a very simple data distribution. New and complex statistical methods based on machine learning have the potential to unravel hidden structures and uncover complex patterns in large database, impossible with traditional statistical methods. Obviously, the ability to unravel underlying complex structures in the data, to provide insights into associations between the different features in the data, to identify sub-populations in the data or they group a set of samples enabling an exploratory observation on subsets of samples, and to produce different inferred functions that maps labeled input data to outcome variables that can be utilized to make predictions of the outcome variables given new input data necessarily change the trial outputs.
Comment 4: an interesting covariate is lacking: has the pt completed his/her RT planned schedule? Any dose reduction? Any drop-out?
Response 4: We thank the Reviewer for this pertinent suggestion. Because ARISE was designed as a prospective cross-sectional study with a single per-patient PMI recorded at the clinical visit (during or at the end of RT), the case-report form did not capture RT adherence variables such as completion of the planned schedule, dose reductions/interruptions, or treatment drop-out. Accordingly, these covariates were not analyzed. We agree they may correlate with pain severity and prescribing, and we now state this explicitly in the Limitations, noting the possibility of residual confounding. At the same time, the cross-sectional design reduces the risk that attrition during RT biased the PMI measured at the visit. We also indicate that future linkage with RT treatment logs would allow testing these hypotheses.
Manuscript changes
- Methods, Study design (added to section 2.2.): “The CRF did not include radiotherapy adherence variables (e.g., completion of the planned schedule, unplanned interruptions/dose reductions, or treatment drop-out); these were therefore not available for analysis.”
- Limitations (added to section 4.3.): “A further limitation is the absence of radiotherapy adherence data. We did not collect information on completion of the planned RT, unplanned interruptions or dose reductions, nor on treatment drop-out. While our cross-sectional design reduces the likelihood that attrition biased the PMI assessed at the clinical visit, these unmeasured variables could still confound the association between patient-/treatment-level factors and analgesic adequacy. Future studies should link clinical assessments with RT treatment logs to evaluate these effects explicitly.”
- Future perspectives (added at the end of the discussion section): “Linkage with radiotherapy delivery records (fractionation, interruptions, completion) would clarify whether adherence metrics influence analgesic adequacy trajectories.”
Comment 5: the performance status, add ECOG, please
Response 5: We thank the Reviewer for this helpful remark. ECOG performance status was already collected and described in the Methods and reported among baseline characteristics. To make this unambiguous in the tables, we have now spelled out the acronym and added the full ECOG definition in the Table 1 footnote.
Manuscript change
- Table 1, legend (added): “ECOG = Eastern Cooperative Oncology Group performance status, 0–4 scale: 0 fully active; 1 restricted in physically strenuous activity but ambulatory and able to carry out light work; 2 ambulatory and capable of all self-care but unable to carry out any work activities; up and about >50% of waking hours; 3 capable of only limited self-care; confined to bed or chair >50% of waking hours; 4 completely disabled.
Comment 6: four CART classifiers, what about to choose and report only one of them? My suggestion is to choose the most representative model and report/comment only that one (of course, including geographical locations and age as covariates/features)
Response 6: We thank the Reviewer for this thoughtful suggestion to streamline the presentation. Our choice to report four CART classifiers was deliberate: each tree captures complementary patterns in the data and illustrates how determinants of analgesic adequacy may vary across clinical contexts. Collapsing to a single “most representative” tree would risk model-selection bias and over-generalization from one specific fit, whereas showing multiple, concise trees enhances transparency and robustness of the exploratory findings. Importantly, age and geographical macro-area were included as candidate features in all CART models. Whether a given tree displays a split on these variables is determined by the data-driven impurity reduction and pruning criteria; absence of a split in a specific tree reflects the algorithm’s selection, not omission of the covariate. We therefore prefer to retain the four classifiers to provide a balanced view of the patterns observed, while keeping the section concise. We are grateful for the Reviewer’s understanding.
Comment 7: figure 2, mind that 34.2% + 74.3% does not add to 100%
Response 7: The probabilities are not addictive. CART algorithm seek predictors and cut points in the predictors that are used to split the sample. The cut points divide the sample into more homogeneous subsamples. The splitting process is repeated on both subsamples. In the present case, this means that in the group of patients having cancer pain (914 patients), 34.2% of them had inadequate analgesia (PMI<0). In the group of patients having non cancer pain (439 patients), 74.3% of them had inadequate analgesia (PMI<0). Therefore, the two probabilities refer to two distinct groups of patients and cannot be addictive.
Comment 8: the same for figures 3,4,5
Response 8: Same explanation as above
Comment 9: table 1, cancer/non-cancer pain does not add to 100%
Response 9: We thank the Reviewer for spotting this rounding inconsistency. We corrected the non-cancer (non-neoplastic) pain percentage from 32.4% → 32.5%, so that cancer vs non-cancer pain now sums to 100.0%. We also added a brief footnote noting that values are rounded to one decimal.
Manuscript change
- Table 1 → “Type of pain” row: replaced: Non-cancer pain: 32.4% with: Non-cancer pain: 32.5%
- Table 1 footnote (added): “Percentages are rounded to one decimal; totals may differ from 100.0% by ±0.1 due to rounding.”
Comment 10: no XAI explainable AI techniques have been used, to convert the black box into a glass one
Response 10: Explainable AI (XAI) refers to methods and techniques that enable humans to understand, interpret, and trust the decisions and outputs of artificial intelligence (AI) models, particularly "black box" machine learning algorithms where the reasoning is otherwise unclear. Then, when we discuss interpretability in machine learning, we refer to the ability to clearly understand and trace how a model reaches its predictions. Interpretable models are those where a human observer can follow the decision-making process and directly link the input features to the output predictions. In contrast, non-interpretable models, often referred to as "black box" models, have a more complex structure, making it challenging to understand the reasoning behind their predictions. A common heuristic for differentiating between these models is as follows:
- Interpretable Models: Models such as decision trees, CART and linear models are considered as intrinsic interpretable. Their structure is designed in a way that each decision or coefficient can be explained and traced back to the input features. They possess a simple, intuitive flowchart-like structure where internal nodes represent decision rules based on feature values, branches denote the outcomes of these decisions, and leaf nodes hold the final predictions. The path from the root to a leaf node provides a clear and understandable decision-making process, which is crucial for explainable AI applications. This is the case of our approach based on CART analysis.
- Non-interpretable Models: Models like neural networks and ensemble methods (e.g, random forests and gradient boosting) are typically non-interpretable. Due to their complexity, consisting of numerous layers, nodes, and parameters, it is not straightforward to trace individual predictions, and they need post hoc explanations as LIME, which stands for Local Interpretable Model-agnostic Explanation, or SHAP, Shapley Additive Explanations (Lundberg, 2017), a method based on cooperative game theory.
Reviewer 2 Report
Comments and Suggestions for Authors
This paper focuses on the possible influence of older age on the management of pain in patients receiving radiotherapy in Italian centers and on age-related factors leading to analgesic undertreatment.
The introduction is comprehensive and presents the clinical relevance of cancer pain management, particularly in elderly patients. The prospective, multicenter, observational design with consecutive patient enrollment across 13 radiotherapy centers is appropriate and robust. The use of modern statistical approaches (LASSO and CART) is methodologically sound and adds value.
The methods are well described, including inclusion/exclusion criteria, pain assessment and adequacy evaluation via the Pain Management Index. Statistical analysis is reported in detail.
The results are presented with clarity and supported by tables and figures: tables are clear and easy to interpret; figures are scientifically correct but visually complex (especially CART trees) and additional explanatory legends could improve the accessibility.
The conclusions are consistent with the data presented. The finding that age alone does not predict undertreatment, while other factors (pain type, RT intent, geographical disparities) are more relevant, is well substantiated.
I suggest:
- Consider adding more recent international references (post-2020) on cancer pain management in elderly populations to strengthen the introduction/discussion.
- Expand figure legends to ensure they can be understood without referring back to the main text.
- A summary table highlighting key predictors of undertreatment stratified by age group could enhance clarity.
After these minor revisions, in my opinion the paper could be accepted.
Author Response
Comment 1: This paper focuses on the possible influence of older age on the management of pain in patients receiving radiotherapy in Italian centers and on age-related factors leading to analgesic undertreatment. The introduction is comprehensive and presents the clinical relevance of cancer pain management, particularly in elderly patients. The prospective, multicenter, observational design with consecutive patient enrollment across 13 radiotherapy centers is appropriate and robust. The use of modern statistical approaches (LASSO and CART) is methodologically sound and adds value. The methods are well described, including inclusion/exclusion criteria, pain assessment and adequacy evaluation via the Pain Management Index. Statistical analysis is reported in detail. The results are presented with clarity and supported by tables and figures: tables are clear and easy to interpret; figures are scientifically correct but visually complex (especially CART trees) and additional explanatory legends could improve the accessibility. The conclusions are consistent with the data presented. The finding that age alone does not predict undertreatment, while other factors (pain type, RT intent, geographical disparities) are more relevant, is well substantiated.
Response 1: We thank the Reviewer for the positive assessment and for noting that the CART outputs would benefit from clearer legends. Because we display CART results as rule tables (rather than tree graphics), we have expanded the legends with a short “how to read” guide that explains how panels relate to parent subgroups, how to interpret n and % PMI < 0, and how to follow a path through the hierarchy. This improves accessibility without altering data or results.
Manuscript change
- Added the following paragraph at the end of each CART figure caption: “Each row is a distinct subgroup defined by the labels shown in that row. When the table has multiple blocks/columns, read left to right: each block further splits the subgroup to its left, and the patient counts within a block sum to their parent. Rows are mutually exclusive and together cover all patients in the parent group. The model considered all candidate predictors listed in Methods; a variable appears only if selected by the algorithm.”
Comment 2: I suggest to consider adding more recent international references (post-2020) on cancer pain management in elderly populations to strengthen the introduction/discussion.
Response 2: We thank the Reviewer for this helpful suggestion. We agree that adding recent, international sources focused on older adults can further strengthen our Introduction and Discussion. We have now cited (i) the 2023 ASCO clinical guidance on opioid use in adults with cancer, (ii) the NCI PDQ (updated 2025) noting specific risks of undertreatment in geriatric patients, (iii) a 2023 JGO report underscoring the need for patient-reported pain in older adults, (iv) a 2024 Drugs & Aging review on geriatric pharmacology considerations, and (v) a 2024 international update on cancer pain management. These additions reinforce our framing that age per se is not the key driver, while geriatrics-specific assessment and organized pain-care processes remain essential.
Manuscript changes
- Introduction section, added: ““In older adults, cancer pain management is further complicated by geriatric syndromes, polypharmacy, and altered pharmacokinetics, which call for systematic screening and individualized, multimodal care. Recent international guidance emphasizes offering opioids when indicated with risk-mitigation strategies and careful dose titration, alongside non-pharmacologic and interventional options, particularly pertinent to older patients. Contemporary reviews and practice resources also highlight that geriatric patients are at heightened risk of undertreatment and benefit from consistent use of patient-reported pain measures. (references added)”
- Discussion section (4.2.), added: “Our finding that age alone was not a determinant of undertreatment aligns with recent international literature, which stresses that geriatric assessment, proactive monitoring, and tailored analgesic choices, not age per se, should guide management in older adults with cancer. Incorporating patient-reported pain assessment and geriatric pharmacology principles may help narrow residual gaps in adequacy. (references added)”
- References list:
Paice JA, Bohlke K, Barton D, Craig DS, El-Jawahri A, Hershman DL, et al. Use of opioids for adults with pain from cancer or cancer treatment: ASCO guideline. J Clin Oncol. 2023;41(4):914-930. doi:10.1200/JCO.22.02198.
PDQ Supportive and Palliative Care Editorial Board. Cancer Pain (PDQ®)–Health Professional Version [Internet]. Bethesda (MD): National Cancer Institute (US); 2025 Apr 24 [updated; cited 2025 Sep 5]. Available from: https://www.cancer.gov/about-cancer/treatment/side-effects/pain/pain-hp-pdq
Arana-Chicas E, Culakova E, Mohamed MR, Tylock R, Wells M, Flannery M, et al. Older adults with advanced cancer report pain not captured by clinician-graded Common Terminology Criteria for Adverse Events (CTCAE). J Geriatr Oncol. 2023;14(3):101480. doi:10.1016/j.jgo.2023.101480.
Pickering G, Kotlińska-Lemieszek A, Krcevski Skvarc N, O’Mahony D, Monacelli F, Knaggs R, et al. Pharmacological pain treatment in older persons. Drugs Aging. 2024;41:959-976. doi:10.1007/s40266-024-01151-8.
Comment 3: I suggest to expand figure legends to ensure they can be understood without referring back to the main text.
Response 3: We thank the Reviewer for this helpful suggestion. As noted in our previous response, we have expanded the legends so each figure/table is self-contained. In particular, the CART outputs (shown as rule tables) now include a brief “how to read” guide.
Comment 4: I suggest a summary table highlighting key predictors of undertreatment stratified by age group could enhance clarity.
Response 4: We thank the Reviewer for this helpful idea. To make the age-dependence immediately clear, we added a very concise, age-stratified summary table that ranks predictors of undertreatment (PMI < 0) as primary / secondary / tertiary (context-dependent) within each age group. This table is descriptive and complements the detailed results.
Manuscript changes (text to paste)
- a) Results — add 1 sentence at the end of the paragraph summarizing determinants
“For readability, we include a concise age-stratified summary table that highlights primary, secondary, and context-dependent predictors of undertreatment (Table 2).”
- b) Supplement (or Main Text) — create the following simple table and caption
Table 2. Age-stratified summary of predictors of undertreatment (PMI < 0).
Ranking reflects consistency/strength across LASSO and CART: Primary = most consistent signals; Secondary = frequent signals; Tertiary = context-dependent (appear in CART subsets). This table is descriptive and does not introduce new estimates.
|
Age group |
Primary |
Secondary |
Tertiary (context-dependent) |
|
18–64 years |
Type of pain |
Aim of therapy |
Type of cancer; Performance status |
|
≥65 years |
Type of pain |
Aim of therapy |
Type of cancer; Location of the RT center; Timing of the visit |
Abbreviation: RT, radiotherapy.
Comment 5: After these minor revisions, in my opinion the paper could be accepted.
Response 5: We sincerely thank the Reviewer for the constructive feedback throughout and for the positive final assessment. We have implemented the suggested minor revisions. We appreciate the Reviewer’s time and consideration
Reviewer 3 Report
Comments and Suggestions for Authors
Interesting, innovative cross-sectional data analysis, however, with many limitations due to the more simplistic data collection and interpretation.
This study subtracted from a categorical analgesic score (0=no analgesics, 1=non-opioid analgesics, 2= weak opioids, and 3=strong opioids) a categorical pain severity score (1=mild pain, NRS 1-4; 2=moderate pain, NRS 5-6; 3=severe pain, NRS 7-10) to receive the Pain Management Index (PMI) which – according tot he authors – should represent untreated pain in case the PMI is negative. The data were obtained from a structured case report form completed only once during a visit to the Department of Radiotherapy. For the data analysis, patients were stratified by age into three groups: young adults (18–35 years), middle-aged adults (36–64 years), and older adults (≥65 years).
Since the authors report a prospective study design and the data from the single case report form were obtained after patients‘ informed consent, the statistics lack a prospective power analysis (patients per age group) with reference to their primary endpoint PMI difference between younger and older patients. In this context, a negative result does not necessarily mean a true negative result, thus, requests a more cautious interpretation of data.
Since the data were obtained only once in a cross-sectional study design, they represent just a snap-shot in time which in the context of radiotherapy is subject to strong fluctuations: depending on the target structure/organ of radiotherapy (this information is lacking), whether it was a curative or a paliative radiotherapy (could this factor be excluded to have an impact), whether it was during or after radiotherapy could this factor be excluded to have an impact), etc. This should be also considered in the interpretation of data.
To interpret a negative PMI score, i.e. the class of analgesics given minus the category of pain severity, is a calculation that is too rough, does not consider efficacy of analgesic treatment, and has not been validated. Accordingly, data interpretation should be more cautious. The authors do not address pain due to radiotherapy which is clinically relevant. What did the authors mean with non-cancer pain; was this pain completely unrelated tot he cancer disease?
The differences between South and North Italy could be due to the much higher number of patients in South Italy and perhaps to a higher number of Departments included (not reported) indicating a higher variability than in North Italy where few but highly specialized Cancer Centers are available.
Apart from their different prediction models , the authors should also give a statistical analysis oft the obtained data so far.
Author Response
Comment 1: Interesting, innovative cross-sectional data analysis, however, with many limitations due to the more simplistic data collection and interpretation.
Response 1: We thank the Reviewer for the positive appraisal of the study’s innovation and for highlighting the limitations related to our pragmatic data collection and the interpretation of cross-sectional findings. The lean case-report form was intentional, to harmonize data across 13 radiotherapy centers, minimize burden, and secure high-quality, consecutive enrollment focused on our primary aim (pain assessment and adequacy via PMI). We fully agree that this design limits granularity (e.g., detailed analgesic dosing/titration, geriatric assessment domains, and RT adherence metrics) and precludes causal inference. To address this, we have further strengthened the Limitations, clarified the rationale for the minimal dataset in Methods, and added to the Conclusions that results are exploratory/hypothesis-generating. We also note that complementary methods (LASSO and CART) were used to enhance transparency, while acknowledging they cannot compensate for unmeasured confounding. We appreciate the Reviewer’s guidance.
Manuscript change
- Methods, Study design/Data collection, added: “To ensure feasibility and consistent data quality across 13 centers with consecutive enrollment, we adopted a minimal, standardized case-report form focused on core variables aligned with the study aim (pain classification and intensity, analgesic regimen to derive PMI, and key RT/tumor descriptors).”
- b) Discussion, Limitations, added: “A key limitation is the pragmatic breadth of our dataset. The CRF did not capture several clinically relevant domains (e.g., detailed analgesic dosing and titration, breakthrough pain, comorbidities/polypharmacy and formal geriatric assessment, or radiotherapy adherence metrics such as interruptions/completion). Together with the cross-sectional design, this limits causal inference and leaves room for residual confounding. While our prespecified modeling strategy (LASSO for variable selection and CART for pattern discovery) improves transparency and reduces overfitting, it cannot compensate for unmeasured variables. Future work should prospectively link clinical assessments with RT treatment logs and richer geriatric/pain measures, with longitudinal follow-up to examine trajectories in analgesic adequacy.”
- Conclusions, added: “However, due to the study limitations, our findings should be interpreted as exploratory and hypothesis-generating, identifying organizational targets for improving pain-care processes rather than implying causality.”
Comment 2: This study subtracted from a categorical analgesic score (0=no analgesics, 1=non-opioid analgesics, 2= weak opioids, and 3=strong opioids) a categorical pain severity score (1=mild pain, NRS 1-4; 2=moderate pain, NRS 5-6; 3=severe pain, NRS 7-10) to receive the Pain Management Index (PMI) which – according to the authors – should represent untreated pain in case the PMI is negative. The data were obtained from a structured case report form completed only once during a visit to the Department of Radiotherapy. For the data analysis, patients were stratified by age into three groups: young adults (18–35 years), middle-aged adults (36–64 years), and older adults (≥65 years).
Response 2: We thank the Reviewer for succinctly summarizing our methods.
Comment 3: Since the authors report a prospective study design and the data from the single case report form were obtained after patients‘ informed consent, the statistics lack a prospective power analysis (patients per age group) with reference to their primary endpoint PMI difference between younger and older patients. In this context, a negative result does not necessarily mean a true negative result, thus, requests a more cautious interpretation of data.
Response 3: We thank the Reviewer for raising this important methodological point. The ARISE study was a pragmatic, multicenter, prospective cross-sectional project with consecutive enrollment over a fixed window; the sample size was determined by feasibility across 13 RT centers, and no a priori power calculation was performed for detecting between-age differences in the PMI. We agree that, in this context, the lack of a detected effect for age does not preclude a small true effect (risk of type II error), and we have now made our interpretation more cautious and explicitly exploratory. To reflect this, we clarify the sample-size rationale in Methods and strengthen the Limitations and Conclusions accordingly.
Manuscript changes
- Methods, Study design, added: “This was a pragmatic, multicenter prospective cross-sectional study with consecutive enrollment over a predefined window. The sample size reflected feasibility across 13 centers; a formal a priori power calculation for between-age differences in the PMI was not performed.”
- Statistical analysis, added: “Analyses were exploratory and estimation-focused, emphasizing variable selection and pattern discovery (LASSO, CART) rather than hypothesis testing for a prespecified age-group contrast.”
- Discussion, Limitations, added: “Another limitation is the absence of a prospective power analysis for detecting age-group differences in analgesic adequacy. Accordingly, the null association for age per se should be interpreted with caution, as small-to-moderate effects cannot be excluded (i.e., potential type II error). These results should be considered hypothesis-generating.”
- Conclusions, added: “These studies should be prospectively powered to detect clinically meaningful between-age differences in PMI are warranted.”
Comment 4: Since the data were obtained only once in a cross-sectional study design, they represent just a snap-shot in time which in the context of radiotherapy is subject to strong fluctuations: depending on the target structure/organ of radiotherapy (this information is lacking), whether it was a curative or a palliative radiotherapy (could this factor be excluded to have an impact), whether it was during or after radiotherapy could this factor be excluded to have an impact), etc. This should be also considered in the interpretation of data.
Response 4: We thank the Reviewer for this important clarification. We agree that, in a radiotherapy setting, pain and prescribing may fluctuate over time, so a single-visit PMI provides a snapshot rather than a trajectory. Our case-report form captured RT intent (curative vs palliative), the timing of the study visit (start vs end of RT), and tumor site, and these factors were considered in our analyses. However, we did not collect granular information on the irradiated target structure/organ or fractionation parameters across centers. We have now made this explicit and emphasize a more cautious interpretation, acknowledging that residual confounding by unmeasured treatment factors cannot be excluded.
Manuscript changes
- Methods, Variables/Data collection, added: “However, we did not collect detailed information on the irradiated target structure/organ, treatment volumes, or fractionation beyond these descriptors.”
- Discussion, Limitations, added: “Because of the cross-sectional design, each patient contributed a single PMI measurement, which offers a snapshot of analgesic adequacy and cannot capture within-patient fluctuations in pain and prescribing during or after RT. Moreover, we lacked granular data on the irradiated target structure/organ and fractionation parameters. Although RT intent and visit timing were recorded and considered in the analyses, residual confounding by unmeasured treatment factors cannot be excluded; therefore, results should be interpreted with appropriate caution.”
Comment 5: To interpret a negative PMI score, i.e. the class of analgesics given minus the category of pain severity, is a calculation that is too rough, does not consider efficacy of analgesic treatment, and has not been validated. Accordingly, data interpretation should be more cautious. The authors do not address pain due to radiotherapy which is clinically relevant. What did the authors mean with non-cancer pain; was this pain completely unrelated to the cancer disease?
Response 5: We thank the Reviewer for these valuable points. The Pain Management Index (PMI) is a widely used, pragmatic indicator of analgesic adequacy in oncology and has been employed across numerous studies and systematic reviews to quantify undertreatment at the population level. At the same time, we agree it is a coarse proxy: it does not incorporate dose, route, adjuvants, timing, or patient response, and therefore a negative PMI indicates potential undertreatment rather than definitive inefficacy. We have now tempered the interpretation accordingly and cite literature both supporting its use and discussing its limitations. Regarding radiotherapy-related pain, we acknowledge its clinical relevance and temporal fluctuations during/after RT; we have clarified how this relates to our pain classification and added citations. Finally, we now define “non-cancer pain” explicitly as pain judged by the treating radiation oncologist not attributable to the cancer (e.g., chronic musculoskeletal-neuropathic conditions), whereas “cancer pain” includes tumor-related pain.
Manuscript changes
- Methods, End Points, added: “The PMI was used as a pragmatic indicator of analgesic adequacy (negative values = potential undertreatment). We recognize that the PMI does not account for dose, co-analgesics, or clinical response; it is intended for population-level quality assessment. Pain classification was recorded by the treating radiation oncologist as: cancer pain (attributable to the tumor) versus non-cancer pain (pre-existing/comorbid pain deemed not attributable to the cancer including radiotherapy-related toxicities).”
- Discussion, Limitations, added: “Therefore, PMI < 0 should be interpreted as potential undertreatment, not proof of ineffective analgesia, and results have to be interpreted with appropriate caution. However, prior work supports its utility for benchmarking while also highlighting its limitations. [new references added]”
- Discussion, Limitations, added: “In radiotherapy settings, pain can fluctuate with target organ, treatment phase, and toxicity, underscoring the need for systematic assessment. Our cross-sectional snapshot captures status at a single visit and cannot represent symptom trajectories during/after RT. [new reference added]”
- References list:
Deandrea S, Montanari M, Moja L, Apolone G. Prevalence of undertreatment in cancer pain: A review of published literature. Ann Oncol. 2008;19(12):1985-1991.
Greco MT, Roberto A, Corli O, Deandrea S, Bandieri E, Cavuto S, et al. Quality of cancer pain management: An update of a systematic review of undertreatment of patients with cancer. J Clin Oncol. 2014;32(36):4149-4154.
Mitera G, Zeiadin N, Kirou-Mauro A, DeAngelis C, Wong J, Sanjeevan T, et al. Retrospective assessment of cancer pain management in an outpatient palliative radiotherapy clinic using the Pain Management Index. J Pain Symptom Manage. 2010;39(2):259-267.
Sakakibara N, Higashi T, Yamashita I, Yoshimoto T, Matoba M. Negative pain management index scores do not necessarily indicate inadequate pain management: A cross-sectional study. BMC Palliat Care. 2018;17(1):102.
Wang K, Tepper JE. Radiation therapy–associated toxicity: Etiology, management, and prevention. CA Cancer J Clin. 2021;71(5):437-454.
Comment 6: The differences between South and North Italy could be due to the much higher number of patients in South Italy and perhaps to a higher number of Departments included (not reported) indicating a higher variability than in North Italy where few but highly specialized Cancer Centers are available.
Response 6: We thank the Reviewer for this important clarification. We agree that unequal cluster sizes (patients and number of Departments) can contribute to macro-area differences. To make this transparent, we now report the distribution of participating Departments and patients by macro-area: North: 3 Departments (including 1 Cancer Center), n=258 patients; Center: 3 Departments (including 1 Cancer Center), n= 157; South: 7 Departments (including 2 Cancer Centers), n=938. Our analyses (LASSO/CART) identify macro-area as a signal at the patient level, but, given the unbalanced number of centers and the absence of multilevel modeling, center-level heterogeneity cannot be excluded as a contributor. We have tempered the interpretation accordingly and indicated that future work should use hierarchical (center-random-effects) models or equal-weight per-center summaries to disentangle macro-area from center effects. We are grateful for the Reviewer’s suggestion.
Manuscript changes
- Methods, Study Design, added: “Thirteen Radiotherapy Departments participated. By macro-area: North = 3 Departments (including 1 Cancer Center), 258 patients; Center = 3 Departments (including 1 Cancer Center), 157 patients; South = 7 Departments (including 2 Cancer Centers), 938 patients.”
- Results, first paragraph, added: “Enrollment was uneven across macro-areas, with 7 Departments in the South contributing 938/1,353 (69.3%) patients, versus 3 Departments each in the North (n=272) and Center (n=168).”
- c) Discussion section 4.1., added: “Moreover, the observed geographic gradient should be interpreted with caution because macro-areas contributed different numbers of Departments and patients, and we did not model center-level clustering. Thus, part of the signal may reflect center heterogeneity rather than geography per se. Future studies should employ multilevel models (center random effects) and/or equal-weight per-center summaries to separate macro-area from center effects.”
Comment 7: Apart from their different prediction models, the authors should also give a statistical analysis of the obtained data so far.
Response 7: Thanks for this suggestion. We added in Table 1 a statistical comparison among the two groups of patients with PMI<0 and PMI ≥0) for all metrics. Categorical data are reported with frequencies and percentage. The Fisher exact test for categorical variables has been used to get the p-values for statistical significance among different groups. This is now inserted in the manuscript.
Reviewer 4 Report
Comments and Suggestions for Authors
Thank you for asking me to review this manuscript reporting adequate pain management in relation to age in a large cohort of patients undergoing radiotherapy in Italy. There are some issues with this paper that require attention before it can be considered suitable for publication:
- The data for this study were derived over 5 years ago, pre-pandemic. We know patient perception and patient care has evolved through this period so this needs to be acknowledged in this paper.
- Young adult was defined as 18-35 years. This is an unconventional age range for young adult; usually defined as 18-25, 18-29 or 18-39.
- We know that the symptom trajectory of patients undergoing RT is a rollercoaster, which could influence their perceptions of pain. Simply classifying patients on treatment and end of treatment is too simplistic. The analysis should correspond with where the patient is in the number of fractions they have received. End of treatment should definitely include whether this is at the peak of symptoms or much later.
- The NRS and analgesic pain scores have been used as primary outcome measures. Details are needed of the reliability and validity of these scores.
- Tables should have the heading at the top and not the bottom of the table.
- The way the data are presented in the figures is confusing and should be presented more clearly.
- The discussion has a lot of repetition about the method – it reads as if the main focus of this study was to use a machine learning method rather than traditional methods of analysis. It would be helpful in the methods for a justification for this method of analysis over and above traditional statistics.
- The table of other studies in the discussion needs to be removed. The discussion should be a compare and contrast of results using the authors interpretation not presenting this in a table for the reader to make this. We also only have the authors opinion that these are the ‘seminal’ pieces as there was no detail of the literature search etc. As this is not a review paper, the ‘seminal’ papers should not be presented as such.
- In the limitations, the strengths of LASSO and CART are presented but there are also a number of limitations that should also be discussed.
- Finally, I think the authors need to provide much greater guidance on how this informs practice. The three operational priorities presented, the first I think evidence is needed to show that the measure they are suggesting is better than the pain management tools currently used in practice, and the other two priorities I would expect to already be in place. It would be helpful to see what guidance the authors could provide specifically related to the predicted factors identified in this study.
Author Response
Thank you for asking me to review this manuscript reporting adequate pain management in relation to age in a large cohort of patients undergoing radiotherapy in Italy. There are some issues with this paper that require attention before it can be considered suitable for publication:
Comment 1: The data for this study were derived over 5 years ago, pre-pandemic. We know patient perception and patient care has evolved through this period so this needs to be acknowledged in this paper.
Response 1: We thank the Reviewer for this important point. Our dataset was collected in late 2019 (pre-COVID-19). We agree that oncology care and patient experience have evolved during and after the pandemic, e.g., wider use of telemedicine, reconfigured radiotherapy pathways (including greater hypofractionation), and service adaptations, which could influence pain assessment workflows and analgesic prescribing, and therefore the external generalizability of our PMI findings. We now explicitly acknowledge this and frame our results as a pre-pandemic baseline that warrants contemporary replication. PMC+1SpringerLink
Manuscript changes
- Methods, 2.2, added: “Data collection occurred in October–November 2019, i.e., prior to the COVID-19 pandemic.”
- Discussion, Limitations, added: “A further limitation is that data were collected pre-pandemic. Since 2020, oncology services have undergone substantial changes, including telemedicine adoption and radiotherapy pathway reconfigurations (e.g., broader use of hypofractionation), with documented effects on patient experience and care processes. These shifts may influence pain assessment and analgesic prescribing; thus, our findings should be interpreted as a pre-COVID baseline, with cautious generalizability to current practice. [new references added]”
- Conclusions, added: “… and they should replicate and update these analyses in post-pandemic cohorts to confirm whether the observed patterns persist under current care models.”
- References list, added:
Garavand A, Khodaveisi T, Aslani N, Hosseiniravandi MH, Shams R, Behmanesh A. Telemedicine in cancer care during COVID-19 pandemic: a systematic mapping study. Health Technol (Berl). 2023;13:665-678.
Piras A, Venuti V, D’Aviero A, Cusumano D, Pergolizzi S, Daidone A, et al. Covid-19 and radiotherapy: a systematic review after 2 years of pandemic. Clin Transl Imaging. 2022;10(6):611-630.
Muka T, Li JJX, Farahani SJ, Ioannidis JPA. An umbrella review of systematic reviews on the impact of the COVID-19 pandemic on cancer prevention and management, and patient needs. eLife. 2023;12:e85679.
Comment 2: Young adult was defined as 18-35 years. This is an unconventional age range for young adult; usually defined as 18-25, 18-29 or 18-39.
Response 2: We thank the Reviewer for noting the variability in “young adult/AYA” definitions. While 15–39 years (or 18–39) is widely used, an 18–35 years definition is also adopted in national AYA programs and peer-reviewed studies, most notably in the Dutch AYA ‘Young & Cancer’ Care Network and related publications. To avoid implying a universal standard, we now refer to our first stratum as “younger adults (18–35 years)” and cite these frameworks. Our key inference, that age per se did not drive undertreatment, does not hinge on the exact boundary.
Manuscript changes
- Methods 2.2, added: “Age was categorized a priori as 18–35, 36–64, and ≥65 years. We refer to 18–35 as ‘younger adults’ for clarity. While Adolescents and Young Adults (AYA) definitions vary internationally (e.g., 15–39 or 18–39 years), an 18–35 framework is also used in national AYA programs and peer-reviewed studies (e.g., the Dutch AYA ‘Young & Cancer’ Care Network), supporting our pragmatic choice. [new references added]”
- References list, added:
Ferrari A, Stark D, Peccatori FA, Fern L, Laurence V, Gaspar N, et al. Adolescents and young adults (AYA) with cancer: a position paper of the ESMO/SIOPE AYA Working Group. ESMO Open. 2021;6(2):100096.
Kaal SEJ, Husson O, van Duivenboden S, Jansen R, Manten-Horst E, Servaes P, et al. Empowerment in adolescents and young adults with cancer: Relationship with health-related quality of life. Cancer. 2017;123(20):4039-4047.
Kaal SEJ, Husson O, van Dartel F, Hermans K, Jansen R, Manten-Horst E, et al. Online support community for adolescents and young adults (AYAs) with cancer: user statistics, evaluation, and content analysis. Patient Prefer Adherence. 2018;12:2615-2622.
Comment 3: We know that the symptom trajectory of patients undergoing RT is a rollercoaster, which could influence their perceptions of pain. Simply classifying patients on treatment and end of treatment is too simplistic. The analysis should correspond with where the patient is in the number of fractions they have received. End of treatment should definitely include whether this is at the peak of symptoms or much later.
Response 3: We thank the Reviewer for this insightful point. We agree that symptoms during RT fluctuate over time and that a single-visit snapshot with a coarse timing variable (“on-treatment” vs “end-of-treatment”) cannot capture fraction number, days since last fraction, or proximity to peak symptoms. To avoid redundancy with earlier comments, we have consolidated this acknowledgment into a single, strengthened paragraph in the Limitations, and we point to the need for future linkage with fractionation logs and serial patient-reported outcomes.
Manuscript changes
- Discussion, Limitations, added: ““Because each patient contributed a single-visit assessment, our data provide a snapshot of analgesic adequacy that cannot capture within-patient fluctuations in pain and prescribing during or shortly after RT. The timing variable (‘on-treatment’ vs ‘end-of-treatment’) is coarse and does not encode the number of fractions received, cumulative dose, days since last fraction, or whether the visit occurred near the peak of acute symptoms. Although RT intent and visit timing were recorded and considered, residual confounding by unmeasured treatment-course factors cannot be excluded; therefore, timing-related contrasts should be interpreted with appropriate caution.”
Comment 4: The NRS and analgesic pain scores have been used as primary outcome measures. Details are needed of the reliability and validity of these scores.
Response 4: We thank the Reviewer for this important point. Our primary endpoint was the Pain Management Index (PMI). The PMI is computed by subtracting the pain-intensity category (derived from the 0–10 Numeric Rating Scale, NRS) from the analgesic class (0 = none; 1 = non-opioid; 2 = weak opioid; 3 = strong opioid). Thus, NRS and analgesic class are components used to derive PMI rather than separate primary outcomes. Regarding measurement properties: the NRS is extensively supported as a valid, reliable, and responsive one-item measure of pain intensity in adults, including oncology populations, with multiple systematic reviews recommending it for clinical and research use. PubMed+2PubMed+2
The analgesic class is not a psychometric scale but an objective coding of the medication(s) in use, aligned with the WHO analgesic ladder framework; as such, inter-rater reliability is expected to be high when drug capture is standardized. Conceptually, it provides face/content validity for PMI as a process indicator of adequacy. We have added concise text and citations to document these points, while also reiterating in the Limitations that neither NRS nor class coding captures dose, route, adjuvants, or clinical response. CNIBGiornale di Etica AMA
Manuscript changes
- Methods 2.4, added: “The NRS is a widely used one-item measure with strong evidence of validity, reliability, and responsiveness in adults (including oncology). [new references added] Analgesic exposure was coded as analgesic class (0 = none; 1 = non-opioid; 2 = weak opioid; 3 = strong opioid), following the WHO analgesic ladder framework; this is an objective classification based on recorded medications and is expected to have high inter-rater reliability under standardized data capture.”
- References list, added:
Hjermstad MJ, Fayers PM, Haugen DF, Caraceni A, Hanks GW, Loge JH, et al. Studies comparing Numerical Rating Scales, Verbal Rating Scales, and Visual Analogue Scales for assessment of pain intensity in adults: a systematic literature review. J Pain Symptom Manage. 2011;41(6):1073-93. doi:10.1016/j.jpainsymman.2010.08.016.
Hawker GA, Mian S, Kendzerska T, French M. Measures of adult pain: Visual Analog Scale for Pain (VAS Pain), Numeric Rating Scale for Pain (NRS Pain), McGill Pain Questionnaire (MPQ), Short-Form McGill Pain Questionnaire (SF-MPQ), Chronic Pain Grade Scale (CPGS), Short Form-36 Bodily Pain Scale (SF-36 BPS), and Measure of Intermittent and Constant Osteoarthritis Pain (ICOAP). Arthritis Care Res (Hoboken). 2011;63(Suppl 11):S240-52.
Karcioglu O, Topacoglu H, Dikme O, Dikme O. A systematic review of the pain scales in adults: Which to use? Am J Emerg Med. 2018;36(4):707-14.
Atisook R, Euasobhon P, Saengsanon A, Jensen MP. Validity and utility of four pain intensity measures for use in international research. J Pain Res. 2021;14:1129-39.
Anekar AA, Hendrix JM, Cascella M. WHO Analgesic Ladder. [Updated 2023 Apr 23]. In: StatPearls [Internet]. Treasure Island (FL): StatPearls Publishing; 2025 Jan–. Available from: https://www.ncbi.nlm.nih.gov/books/NBK554435/ Centro Biotech
Comment 5: Tables should have the heading at the top and not the bottom of the table.
Response 5: Thank you. We corrected the position of the headings as suggested.
Comment 6: The way the data are presented in the figures is confusing and should be presented more clearly.
Response 6: We thank the Reviewer for this helpful observation. As detailed in our earlier replies, we have already improved clarity by expanding all CART (tabular) legends with a concise “how to read” guide.
Manuscript change
- Added the following paragraph at the end of each CART figure caption: “Each row is a distinct subgroup defined by the labels shown in that row. When the table has multiple blocks/columns, read left to right: each block further splits the subgroup to its left, and the patient counts within a block sum to their parent. Rows are mutually exclusive and together cover all patients in the parent group. The model considered all candidate predictors listed in Methods; a variable appears only if selected by the algorithm.”
Comment 7: The discussion has a lot of repetition about the method – it reads as if the main focus of this study was to use a machine learning method rather than traditional methods of analysis. It would be helpful in the methods for a justification for this method of analysis over and above traditional statistics.
Response 7: Thanks for this suggestion. We now added in the method section an explanation of using machine learning over traditional statistics. “In addition to conventional statistical analysis, because new and complex statistical methods based on machine learning have the potential to unravel hidden structures and uncover complex patterns in large database, we performed the following supervised ma-chine learning approach.”
Comment 8: The table of other studies in the discussion needs to be removed. The discussion should be a compare and contrast of results using the authors interpretation not presenting this in a table for the reader to make this. We also only have the authors opinion that these are the ‘seminal’ pieces as there was no detail of the literature search etc. As this is not a review paper, the ‘seminal’ papers should not be presented as such.
Response 8: We thank the Reviewer for this clear guidance. We agree that, as this is not a review, the Discussion should present our interpretation rather than delegate comparison to a table. We have therefore removed the table of other studies, eliminated the term “seminal”, and replaced the section with a concise narrative compare–contrast that cites a few illustrative (not exhaustive) references. We also clarify that we did not perform a formal literature search.
Manuscript changes
- (previous) Table 2 has been removed)
- Discussion, 4.2: the term “benchmark studies” has been removed
- Discussion, 4.2: An interpretation (with three different motivations) of the difference between the results of our study and previous ones is already present in the discussion (4.2). However, we have also added to that section a reference to more recent studies, as follows: “Our finding that age alone was not a determinant of undertreatment aligns with recent international literature, which stresses that geriatric assessment, proactive monitoring, and tailored analgesic choices, not age per se, may guide management in older adults with cancer. Incorporating patient-reported pain assessment and geriatric pharmacology principles may help narrow residual gaps in adequacy”
- References list: added:
Paice JA, Bohlke K, Barton D, Craig DS, El-Jawahri A, Hershman DL, et al. Use of opioids for adults with pain from cancer or cancer treatment: ASCO guideline. J Clin Oncol. 2023;41(4):914-930.
PDQ Supportive and Palliative Care Editorial Board. Cancer Pain (PDQ®)–Health Professional Version [Internet]. Bethesda (MD): National Cancer Institute (US); 2025 Apr 24 [updated; cited 2025 Sep 5]. Available from: https://www.cancer.gov/about-cancer/treatment/side-effects/pain/pain-hp-pdq
Arana-Chicas E, Culakova E, Mohamed MR, Tylock R, Wells M, Flannery M, et al. Older adults with advanced cancer report pain not captured by clinician-graded Common Terminology Criteria for Adverse Events (CTCAE). J Geriatr Oncol. 2023;14(3):101480. .
Pickering G, Kotlińska-Lemieszek A, Krcevski Skvarc N, O’Mahony D, Monacelli F, Knaggs R, et al. Pharmacological pain treatment in older persons. Drugs Aging. 2024;41:959-976.
Comment 9: In the limitations, the strengths of LASSO and CART are presented but there are also a number of limitations that should also be discussed.
Response 9: Thanks for this suggestion. Now we added the following sentence in the limitations section: “Lastly, also Lasso regression presents some limitations, in particular its difficulty with highly correlated features, where it arbitrarily selects one predictor and excludes others, potentially leading to unstable or biased results. It can also produce a biased estimator because the penalty can shrink coefficients towards zero, causing the model to underfit the data if the penalty is too high. Furthermore, Lasso can be problematic for non-sparse datasets with many relevant features, as it will select only a subset of predictors by forcing many coefficients to zero.”
Comment 10: Finally, I think the authors need to provide much greater guidance on how this informs practice. The three operational priorities presented, the first I think evidence is needed to show that the measure they are suggesting is better than the pain management tools currently used in practice, and the other two priorities I would expect to already be in place. It would be helpful to see what guidance the authors could provide specifically related to the predicted factors identified in this study.
Response 10:
We thank the Reviewer for requesting more concrete, practice-oriented guidance. Rather than proposing a new tool, we clarify how to operationalize what is already recommended, and we ground the guidance in our predictors (non-cancer pain, curative-intent pathways, center-level variation).
Manuscript changes
- Discussion, 4.4, integrated new paragraph: “Three operational priorities emerge. First, routine documentation of the PMI alongside pain intensity would enable real-time identification of undertreated individuals. In Italy, systematic pain recording is mandated by national policy (Italian low 38/2010), but documentation alone does not always trigger timely action; using PMI < 0 as a simple “adequacy check” functions as a screen for potential undertreatment and prompts same-day review/titration complementing, not replacing, existing pain tools. Second, targeted education for physicians and nursing staff, covering opioid titration, adjuvant use, and communication skills, could enhance both detection and management of pain. While such training is expected in routine care, evidence shows persistent knowledge gaps and variable guideline adherence, particularly in radiotherapy departments; brief refreshers (e.g., titration checklists, short e-learning) can close these gaps. [new reference added]. Third, symptom control should be embedded within multidisciplinary pathways that integrate oncologists, palliative care specialists, pharmacists and psychologists. Although multidisciplinary team pathways often exist on paper, several studies report limited bidirectional referral between radiotherapy and palliative care; pragmatic steps include appointing a pain/palliative liaison, setting automatic referral triggers (e.g., persistent PMI < 0 at two visits), and auditing undertreatment rates by center to reduce unwarranted variation. [new reference added]. Given our predictors, these actions should be prioritized for patients with non-cancer pain, those on curative-intent pathways, and in Center/South macro-areas.”
- References list, added:
Scirocco E, Cellini F, Donati CM, Capuccini J, Rossi R, Buwenge M, Montanari L, Maltoni M, Morganti AG. Improving the Integration between Palliative Radiotherapy and Supportive Care: A Narrative Review. Curr Oncol. 2022 Oct 19;29(10):7932-7942.
Round 2
Reviewer 1 Report
Comments and Suggestions for Authors
The Authors were able to fully solve the previous concerns, congrats!
Author Response
Comment 1: The Authors were able to fully solve the previous concerns, congrats!
Response 1: Thank you for your comments and your final review.
Reviewer 3 Report
Comments and Suggestions for Authors
no more comments
Author Response
Comment 1: no more comments
Response 2: thank you for your comments and help.
Reviewer 4 Report
Comments and Suggestions for Authors
Thank you for responding to my comments. These have been mostly addressed however, figure 2-5, the additional text is helpful but the final row contains the proportion for the block above. These numbers need to be moved up and align with the boxes they refer to.
Author Response
Comment 1: Thank you for responding to my comments. These have been mostly addressed however, figure 2-5, the additional text is helpful but the final row contains the proportion for the block above. These numbers need to be moved up and align with the boxes they refer to.
Response 1: Thanks for the advice. The figures are actually clearer this way.